# Data-Efficient Challenges in Visual Inductive Priors: A Retrospective

## Abstract

Deep Learning requires large amounts of data to train models that work well. In data-deficient settings, performance can be degraded. We investigate which Deep Learning methods benefit training models in a data-deficient setting, by organizing the "VIPriors: Visual Inductive Priors for Data-Efficient Deep Learning" workshop series, featuring four editions of data-impaired challenges. These challenges address the problem of training deep learning models for computer vision tasks with limited data. Participants are limited to training models from scratch using a low number of training samples and are not allowed to use any form of transfer learning. We aim to stimulate the development of novel approaches that incorporate prior knowledge to improve the data efficiency of deep learning models. Successful challenge entries make use of large model ensembles that mix Transformers and CNNs, as well as heavy data augmentation. Novel prior knowledge-based methods contribute to success in some entries.

## 1 Introduction

Deep learning more and more depends on availability of large-scale training datasets. However, the cost of collecting and labeling such datasets scales with their size. Even if the issue of costly labeling can be avoided, training with large unlabeled datasets still uses large amounts of energy, contributing to carbon emissions (Strubell et al., 2020; Schwartz et al., 2020). Furthermore, datasets and compute at such scale is limited to a few powerful big tech companies. Additionally, data at such a scale may not be available for some domains at all. The Visual Inductive Priors for Data-Efficient Deep Learning workshop (VIPriors) therefore encourages research in learning from small datasets, by way of combining the learning power of deep learning with hard-won prior knowledge from specific domains.

The VIPriors workshop ran for four editions at ICCV and ECCV from 2020 to 2023. Aside from featuring a paper track, each workshop hosts multiple challenges, where competitors train computer vision models on small training datasets, challenging them to find competitive solutions without the large quantities of data that power state-of-the-art deep computer vision models. The chosen tasks and datasets are tailored to be data-deficient and, where we are able to use custom datasets, contain prior knowledge that competitors can incorporate in their solutions. As far as we know, our VIPriors challenges are the only challenges that evaluate models in a data-deficient setting without allowing domain adaptation methods.

In this work, we describe all four editions of the VIPriors challenges. Over the course of four years of challenges, we organize challenges on five distinct computer vision tasks: image classification, object detection, segmentation, action recognition and re-identification. For each task, we describe the challenge setup over the years, the accumulated results of all challenges and the most notable methods used by competitors.

We find that large model ensembles and heavy use of data augmentation are important factors in successful entries. Successful entries also mix Transformer-type architectures with CNN-type architectures in their ensembles. Even though a secondary aim of our challenge is for novel prior knowledge-based methods to be successful in a data-deficient setting, only a few such entries use prior knowledge to success.

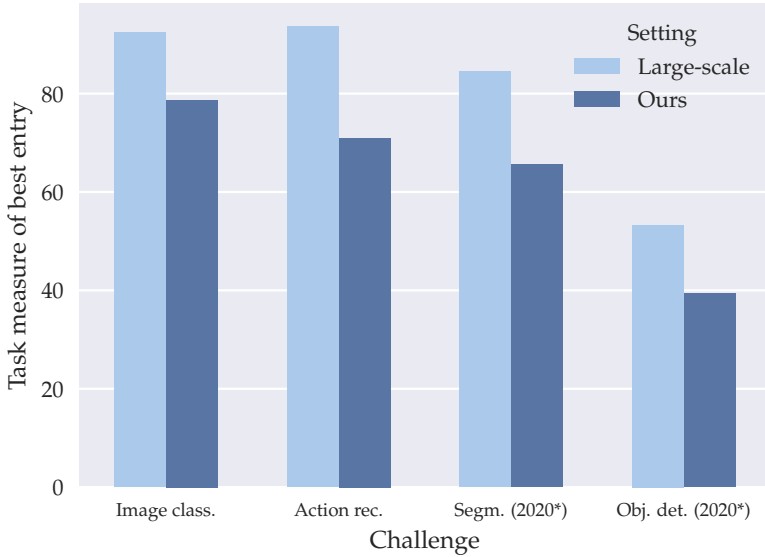

Figure 1: Comparing the winning entries in our challenges against those in corresponding large-scale benchmarks. Performance on our data-deficient settings are significantly worse, highlighting the need for research on data-deficient computer vision. For details, see Section 6.

## 2 Related Works

**Related tasks.** Our data-deficient setting is a supervised training setting with limited labeled samples available. There are other tasks that similarly aim to enable deep learning with little to no data. The category of n-shot learning tasks is characterized by providing the learner with only $n$ training samples, where $n$ can be zero (zero-shot learning), one (one-shot learning) or a few samples (few-shot learning) (Kundu; Wang et al., 2020). To solve this problem, learners often have to resolve to using additional information, for example by transfer learning from another training distribution.

In contrast, in our challenges we provide more data than used in few-shot learning, but much less than in common supervised training benchmarks. Where n-shot learning tasks often require alternative learning or transfer methods to perform, our data-deficient tasks can be approached with regular supervised learning, though performance may not be optimal. Our challenges explore the degree to which supervised learning struggles with this little amount of data and to which extent prior knowledge can alleviate these problems.

**Related datasets.** Several ImageNet-derivatives exist for data-deficient image classification. Mini-ImageNet (Vinyals et al., 2016) was proposed for evaluating few-shot methods and is intended to be harder than CIFAR-10 but easier than full ImageNet. It contains 600 samples per class at full ImageNet image size. TinyImageNet (Ali) was created by the organizers of the CS231n course at Stanford University to serve as benchmark for a course project. It contains 500 samples per class, downsampled to a resolution of 64 pixels squared. Other unnamed subsets of ImageNet have also been proposed (Brigato et al., 2021).

In a similar vein, we use subsampling of ImageNet to create small training datasets for our image classification challenge. In particular, we only use 50 samples per class, and in contrast to TinyImageNet we do not downsample the images.

**Related challenges.** The *Large-Scale Few-Shot Learning Challenge*[1] was hosted at ICCV 2019. As the name suggests, it is a few-shot learning challenge, with five training samples per class. In contrast, our data-deficient setting provides more samples per class.

---

[1] https://lsfsl.net/cl/

The *Cross-Domain Few-Shot Learning Challenge*[2] (CD-FSL) was run at CVPR 2020. A continuation of this challenge was run at ICCV 2021 as the classification track of the *Learning from Limited and Imperfect Data Challenge*[3] (L2ID). The target datasets to solve are data-deficient datasets such as EuroSAT and ISIC, with a similar data scale to our challenges. However, in addition to learning from limited data, domain adaptation from ImageNet was encouraged. In contrast, we do not allow domain adaptation methods in our challenges, rather focusing on learning models from scratch with limited data.

As far as we know, our VIPriors challenges are the only challenges that evaluate models in a data-deficient setting without allowing domain adaptation methods.

## 3 Challenges

Throughout the course of four editions of challenges, we have hosted five different computer vision tasks as challenges, with a varying number of tasks hosted each year. Common cause between all challenges is that the number of training samples are reduced to a small number. Where we adapt existing datasets we choose random subsets of the available data. In other cases, we adapt private datasets acquired through collaboration with industry, which are already so small to be appropriate for the challenge.

We choose to use small training datasets to realistically reproduce the setting of working in a data-deficient setting. Another goal of our challenges is to encourage the solutions to use visual prior knowledge of each task. To this end, we impose further rules to attempt to rule out usage of alternative methods that alleviate data scarcity. These rules are:

- Models shall be trained from scratch with only the given dataset.
- The usage of other data rather than the provided training data, such as pre-training the models on other data and transfer learning methods, are prohibited. It is however allowed to train with synthetic data generated from the training data.

### 3.1 Image classification

Image classification is a cornerstone task in computer vision for its practical use and simple requirement of predicting only a single label. Architectures designed for many other tasks use image classification networks as backbones (Ren et al., 2016; He et al., 2017; Cai & Vasconcelos, 2018b; Chen et al., 2019a). Innovations in image classification networks therefore have ripple effects throughout the whole field of computer vision. Therefore, we include this task in our challenges.

For the dataset we use a subset of ImageNet (Deng et al., 2009). ImageNet has been the de facto benchmark standard for image classification since its inception (Russakovsky et al., 2015). Models trained on ImageNet, even with a reduced number of samples, transfer well to other tasks and distributions (Huh et al., 2016). This motivates us to use ImageNet for our challenges, expecting that improvements in training models for a data-deficient ImageNet will translate to other tasks.

Our subset of ImageNet is a random sample of fifty images from 1,000 classes for training, validation and testing. This is more samples per class than for a typical few-shot setting, but much less than for a typical supervised learning setting. We evaluate models on top-1 accuracy.

This challenge ran from 2020 until 2022. We did not run this challenge for the final edition in 2023, as we felt that competitor interest for challenges had shifted from image classification to more applied tasks like object detection and instance segmentation.

### 3.2 Object detection

Object detection is a widely applied task in computer vision. Its requirement to predict bounding boxes has driven researchers to invent widely varying architectures over the last decade, from ROI-based detec-

---

[2]https://www.learning-with-limited-labels.com/challenge
[3]https://l2id.github.io/challenge_classification.html

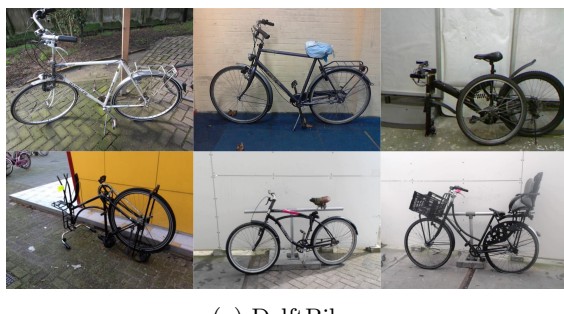

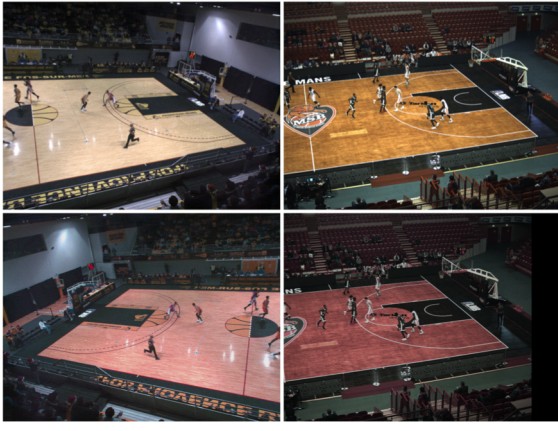

(a) DelftBikes

(b) SynergySports Basketball

Figure 2: Example images from the a) DelftBikes and b) SynergySports Basketball dataset.

tors (Girshick et al., 2014; Ren et al., 2016; He et al., 2017) to YOLO-like dense predictors (Redmon et al., 2016) and recently Transformers such as DETR (Carion et al., 2020). For its widespread applied use, we include this task in our challenges.

We use the DelftBikes (Kayhan et al., 2021) dataset for our object detection challenge. The datasets contains 10,000 images of bikes, photographed from the side (Fig. 2a). Each image contains labels for 22 different bike parts that are annotated with bounding box, class and object state labels. We evaluate models using the same measure as used in the COCO benchmark (COC): Average Precision @ 0.50 : 0.95 (AP).

This dataset is larger than other datasets used in our challenges, but still much smaller than other benchmark datasets for object detection such as COCO. We use this dataset for the strong consistency in bike pose and location of bike parts, which enables competitors to use prior knowledge in their solutions.

This challenge ran from 2021 until 2023. For the 2020 edition, we ran the same challenge but used a subset of approximately 6,000 images from COCO (COC).

### 3.3 Segmentation

Segmentation is a similar task to object detection, with the difference that instead of bounding boxes object instances are segmented by predicting pixel-wise masks. As masks can capture object outlines more precisely than bounding boxes, segmentation has become a popular benchmark in recent years (COC). We therefore include a segmentation task in our challenges.

Within the group of segmentation tasks, we distinguish semantic segmentation, where a single mask per category is predicted, and instance segmentation, where a mask is predicted for each object instance in the image. For our challenge, we use an instance segmentation dataset kindly provided by SynergySports[4], who collaborate with us on organizing this challenge. This dataset contains 18,232 still images taken from recordings of basketball games, where the objective is to segment basketball players and the ball. See Figure 2b for example images. We evaluate models using Average Precision @ 0.50 : 0.95 (AP).

We use this custom dataset for the consistency in camera pose and object appearances that enable competitors to use prior knowledge in their solutions.

This challenge ran from 2021 until 2023. For the 2020 edition, we instead hosted a semantic segmentation task, using a subset of Cityscapes (Cordts et al., 2016) called MiniCity. This subset consists of a train, validation and test set of 200, 100 and 200 images, respectively.

---

[4]https://synergysports.com

### 3.4 Action recognition

Video is the next frontier for computer vision. Action recognition is the task of classifying actions in video clips. Many action recognition are deep, heavy models that need to learn from a lot of data (Carreira & Zisserman, 2017; Feichtenhofer et al., 2019). We aim for our data-deficient action recognition challenge to further research into less data-hungry action recognition models.

To make a dataset for this challenge, we adapt the Kinetics400 dataset (Kay et al., 2017), the de facto benchmark for action recognition, into *Kinetics400VIPriors* by taking a subset. Our training set consists of approximately 100 clips per class (40,000 clips total), while the validation and test sets contain about 25 and 50 clips per class, respectively. For evaluation, we use the average classification accuracy across all classes on the test set.

This challenge ran in 2021 and 2022. For the 2020 edition, we instead used the UCF101 dataset (Soomro et al., 2012), which in itself is already quite small. However, after the results of the 2020 challenge we felt that the high accuracies achieved showed that the competitors were not challenged enough with this dataset. After 2022, we chose to not run this challenge for the final edition, as we wanted to focus our efforts and this challenge did not receive as much interest from competitors as the other challenges.

### 3.5 Re-identification

Re-identification is the task of accurately retrieving the right person from an unseen gallery of photos given an unseen query photo. Typically, models learn to embed pictures of a training set of persons into vector embeddings to represent identities. At test time, the model is then presented with a query photo, which needs to match one of the unseen gallery images by matching the vector embedding. Having access to enough training identities is important. We therefore want to challenge competitors to achieve the same performance in a data-deficient setting.

We collaborated with SynergySports on creating a new, small dataset for this challenge. The provided dataset contains 954 identities and 18,232 images, taken from frame sequences of recordings of basketball games, similar to the data source for the Segmentation challenge (see Sec. 3.3 and Fig. 2b). The dataset is split in training and testing identities. For the validation and test sets, each sequence is truncated to twenty frames, where the first frame is used as query image, and the other frames are used as gallery images. We use top-1 accuracy to evaluate models.

We use this custom dataset for the unique poses and appearances of the persons in it, namely players and referees in a basketball game. We expect that the specific and consistent appearance of this dataset enables competitors to use prior knowledge in their solutions.

This challenge only ran in 2021, where the winners of the challenge achieved a very high score. Because this showed that this setting was not challenging enough, we decided to not run this challenge again moving forward.

## 4 Submission and evaluation

Our challenges are published on the CodaLab platform[5] (Pavao et al., 2023), the latter of which also hosts the technical components of each competition. We make all training and validation data, as well as tooling and baseline models, available through CodaLab and a GitHub toolkit.[6]

To submit to the challenge, competitors are required to register with CodaLab, generate their models predictions over the test set on their own hardware, and upload these to CodaLab in a provided format. The CodaLab platform then automatically computes the evaluation measures and composed online rankings. The labels of the test set are withheld from competitors. As a measure to prevent overfitting to the test set, we limit the number of times an entry can be evaluated on the test set on CodaLab. We note some

---

[5]2020-2021: `https://competitions.codalab.org/`,
2022-2023: `https://codalab.lisn.upsaclay.fr/`
[6]`https://github.com/VIPriors/vipriors-challenges-toolkit`

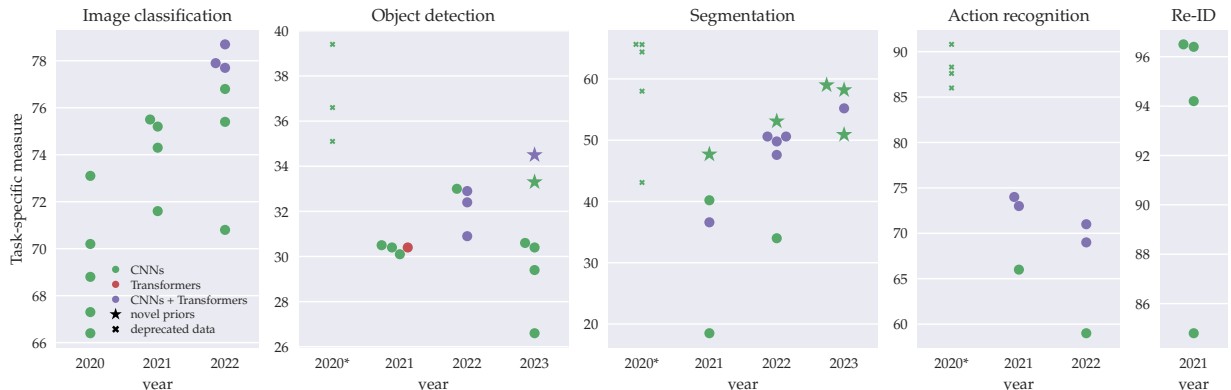

Figure 3: Results of all challenges. Most challenges show improvement in the best submissions year by year. Hybrid architectures are prevalent under well-performing submissions. Each dot represents an entry in the final rankings. The 2020 edition of the object detection, segmentation and action recognition challenges used deprecated datasets with respect to the other years of the same challenge. We indicate what type of architectures were used as well as which entries used some novel methodology inspired by prior knowlegde on the task.

evidence suggesting that competitors evade these evaluation limits by registering with multiple accounts. As we cannot prove this, we do not address this.

At the time of each challenge closing, we compose official rankings by retaining only qualifying entries. To qualify, an entry has to be accompanied by a tech report, either published on the internet or submitted to the organizers by a given later date. We use these reports to verify the validity of the entry, and to enable our analysis of the methodology of each entry in the challenge reports. The final rankings are then published at the time of the live workshop program on the website and in the challenge report. The full results over all editions of each challenge, including details on the methodology used, are given in the appendix (Tables 2–6). We visualize the results of all challenges in Figure 3.

## 5 Methods

Analysis of the methods used by competitors shows patterns within individual challenges as well as patterns that generalize over all challenges. This section discusses these patterns and what we can conclude from them.

**Model ensembling.** Figure 4 shows that many successful entries use large model ensembles. We speculate that the ease of use of model ensembling plays a role in competitors choosing ensembles: many architectures and backbones are available in various libraries, making it easy to plug in an extra model. Ensembles may also cover a wider range of inductive priors than a single model, making the ensemble more successful on a data-deficient task than a single model.

**Architectures.** As to the architectures used by competitors, our challenges experienced the advent of Transformers firsthand. Figure 3 shows that Transformers start to see use in our challenges from 2021 onwards. We find that they are initially only used sporadically as backbones to a CNN-type architecture, but then increasingly as separate architectures, often combined with other inductive priors such as CNN-type architectures in ensembles.

**Data augmentation.** Heavy use of data augmentation is apparent among successful entries, as shown in Figure 5. The particular data augmentations differ per challenge, but the volume of augmentations is consistent. We expect data augmentations are successful in our challenges because they are an easy way to artificially increase the number of training samples in a way that uses prior knowledge on the task at hand. As for the popularity of data augmentations, we expect their ease of use may be a factor.

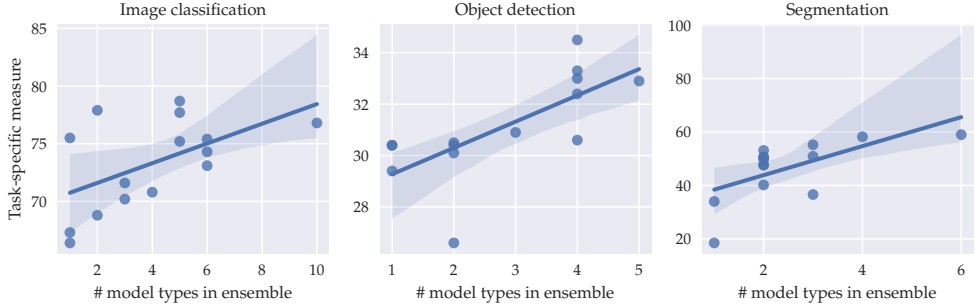

Figure 4: Results of selected challenges plotted against the number of distinct backbone types used in each entry. Each challenge shows a correlation where larger ensemble size correlates with better performance. Each dot represents an entry in the final rankings. Lines represent regression model fits performed by Seaborn (Waskom, 2021). We exclude the action recognition and re-identification challenges as they have too little entries to analyze.

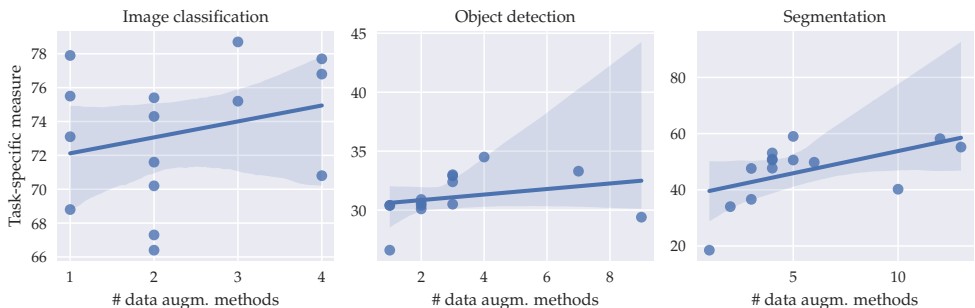

Figure 5: Results of selected challenges plotted against the number of data augmentations used in each entry. Each challenge shows a correlation where larger number of augmentation methods correlates with better performance. Each dot represents an entry in the final rankings. Lines represent regression model fits performed by Seaborn (Waskom, 2021). We exclude the action recognition and re-identification challenges as they have too little entries to analyze.

**On prior knowledge in entries.** Interestingly, we did not see as many novel prior-based methods being implemented in successful entries as we had expected. There are some cases where a novel method may have contributed to success, but only in combination with other factors like ensembling and data augmentation. We speculate that inventing a novel prior-based method is more strenuous on competitors, making them instead choose easier to implement methods such as ensembles and data augmentations.

Outside of novel methodological contributions, prior knowledge was used through pre-existing prior-based methods. One notable example is the use of BlurPool (Zhang, 2019), which specifically enhances translation equivariance in CNNs. We posit that the extensive use of data augmentation policies can also be counted as a use of prior knowledge.

## 5.1 Image classification

In the entries of the image classification task, we note some specific trends. First of all, we confirm that the general trends of large model ensembles and heavy use of data augmentation hold for this task. Specifically, AutoAugment (Cubuk et al., 2018) and CutMix (Yun et al., 2019) are consistently used by almost all competitors. Transformers (specifically ConvNeXt (Liu et al., 2022)) appear in this task in 2022, and contribute to the top three entries in that year. Popular methods to boost performance include label smoothing and some form of extra training on hard or resized samples. Several entries use alternative or

additional loss functions such as focal loss and ArcFace (Deng et al., 2019) loss. Almost all entries use regular supervised training, with a couple entries using some form of knowledge distillation without using additional data. No entries introduce any novel prior-based methodology. However, winning entries in 2021 and 2022 use BlurPool (Zhang, 2019) to great success.

**Winning teams and methods.** Ma et al. from Xidian University are the winners of this challenge, scoring a top-1 accuracy of 78.7. They ensemble of five types of models, including both CNNs and Transformers. They apply various data augmentations to make the data more diverse. Additionally, they apply several optimization tricks, such as patching, hard fusion and random image cropping.

## 5.2 Object detection

In object detection, most entries use between two and four distinct backbone types in ensembles. Again, there is strong evidence for the benefit of large model ensembles. Many entries combine CNNs such as Cascade RCNN (Cai & Vasconcelos, 2018b) or YOLO (Redmon et al., 2016) with Transformers such as Swin (Liu et al., 2021a) and ConvNeXt (Liu et al., 2022). YOLO becomes the dominant architecture starting with version seven and eight of the YOLO framework (Wang et al., 2022; git). Correspondingly, the data augmentations implementations coupled with YOLOv8 are popular, as well as those implemented in the Albumentations library (Buslaev et al., 2020a). Specific augmentations used often are AutoAugment (Cubuk et al., 2018), Mosaic (Bochkovskiy et al., 2020), MixUp (Zhang et al., 2018) and some form of multi-scale augmentation. Ensembles are combined with Soft NMS (Bodla et al., 2017), Weighted Boxes Fusion (Roman Solovyev & Gabruseva, 2021) or Model Soups (Wortsman et al., 2022) to combine predictions.

Overall, there are little entries that use novel prior-based methods. The exception is the 2023 edition, where the top two winning methods propose some novel prior-based methods.

**Winning teams and methods.** Zhao et al. from the Vision Intelligence Department of Meituan are the winners of this challenge, scoring an AP of 34.5. They use a Cascade RCNN (Cai & Vasconcelos, 2018b) with a ConvNeXtV2 backbone (Woo et al., 2023). They contribute a synthetic dataset created from horizontal and vertical recombinations of binary pairs of samples, on which they pre-train. Furthermore, they retrain the model on manually identified hard classes and apply SWA (Izmailov et al., 2018b).

## 5.3 Segmentation

For segmentation, the same patterns apply when it comes to the number of models in ensembles and data augmentations. CutMix (Yun et al., 2019), MixUp (Zhang et al., 2018), RandAugment (Cubuk et al., 2020a), Moasic (Bochkovskiy et al., 2020) and CopyPaste (Ghiasi et al., 2021) are used by many entries, with CopyPaste being unique to this task. RCNN-type architectures (Cascade RCNN (Cai & Vasconcelos, 2018b), Mask RCNN (He et al., 2017), then later on HTC (Chen et al., 2019b)) dominate in this task. Where Transformers are used, they are almost exclusively used as backbone networks in an RCNN-type architecture. In the early editions of this challenge, backbones used are mostly HRNet (Wang et al., 2019b) and CBSwin-T (Liang et al., 2021b), while by 2023 there is a large variety of backbone networks, including anything from HTC, CBSwin, Mask2Former (Cheng et al., 2022a), (Carion et al., 2020), ViT-Adapter (Chen et al., 2022) to Mask RCNN with FPN (Lin et al., 2017a). The best entries use (a variant of) Stochastic Weight Averaging (Zhang et al., 2020b) to ensemble multiple instances of the same model. Regular ensembling is also used through Weighted Boxes Fusion and Model Soups.

Interestingly, each edition of this challenge was won by an entry proposing a novel prior-based method.

**Winning teams and methods.** Zhang et al. from Xidian University are the winners of this challenge, scoring an AP of 59.0. They propose a novel method called *Orthogonal Uncertainty Representation* (OUR), which broadens the geometric manifold (Ma et al., 2023) of underrepresented classes. Furthermore, they use multiple backbones with a Mask RCNN (He et al., 2017) model and a Seesaw loss (Wang et al., 2021d).

Table 1: Statistics and winners of each challenge.

| Challenge | Entries | Qualified entries | Winning team | Winning score |
|---|---|---|---|---|
| Image classification | 33 | 15 | Tianzhi Ma, Zihan Gao, Wenxin He, Licheng Jiao *School of Artificial Intelligence, Xidian University.* | 78.7 |
| Object detection | 87 | 17 | Jiawei Zhao, Xuede Li, Xingyue Chen, Junfeng Luo. *Vision Intelligence Department (VID), Meituan.* | 34.5 |
| Segmentation | 128 | 18 | Junpei Zhang, Kexin Zhang, Rui Peng, Licheng Jiao, Fang Liu, Lingling Li, Yuting Yang. *Xidian University, Xi'an, Shaanxi.* | 59.0 |
| Action recognition | 25 | 10 | Ishan Dave, Naman Biyani, Brandon Clark, Rohit Gupta, Yogesh Rawat and Mubarak Shah. *Center for Research in Computer Vision (CRCV), University of Central Florida.* | 74 |
| Re-identification | 12 | 4 | Cen Liu, Yunbo Peng, Yue Lin. *NetEase Games AI Lab.* | 96.5 |

### 5.4 Action recognition

For the task of action recognition, we have less entries to analyze. It is therefore hard to confirm the general trends on large model ensembles and heavy use of data augmentations for this task. On KineticsVIPriors, only ensembles are used, which combine two-stream methods with Transformer methods. Interestingly, the data augmentations used are the same as for appearance-only tasks (namely CutMix (Yun et al., 2019), MixUp (Zhang et al., 2018), AutoAugment (Cubuk et al., 2018)). The only entry that makes video versions of these augmentations places last in the 2020 edition of this challenge. Outside of this entry, there are very little attempts to customize solutions to the priors of this task.

**Winning teams and methods.** Dave et al. from the University of Central Florida are the winners of this challenge with an average accuracy of 74%. They propose to combine several state-of-the-art methods that have shown promising results in data-deficient settings. They use convolutional (R3D(Hara et al., 2018) and I3D(Carreira & Zisserman, 2017)) as well as attention-based (MViT(Fan et al., 2021)) models. Self-supervised pre-training (TCLR(Dave et al., 2021)) is applied to the convolutional methods, before they are fine-tuned with both RGB and optical flow frames. The Transformer-type model (MViT) is trained using only RGB frames.

### 5.5 Re-identification

As this challenge ran only for one year, with only four qualifying entries in the final ranking, we cannot draw any strong conclusions from this data. In fact, the entries are very similar in chosen methodology, which could indicate that the our sample of competitors is biased in some way. We do note that ensembles used only contain ResNets (He et al., 2015) and ResNet derivates (Zhang et al., 2020a; Xie et al., 2017). The most popular augmentation methods are Random Erasing (Zhong et al., 2020), color jitter, random flipping and AutoAugment (Cubuk et al., 2018). All entries use at least triplet loss (Weinberger & Saul, 2009) and circle loss (Sun et al., 2020b). Re-ranking (Zhong et al., 2017) is used in the top three out of four entries. None of the entries use any form of novel prior-based methodology.

**Winning teams and methods.** Liu et al. from NetEase Games AI Lab are the winners of this challenge with a top-1 accuracy of 96.5%. They use online difficult sample mining, in particular an algorithm similar to (Shrivastava et al., 2016), so as to remove the hard, occluded annotations. They divide occluded annotations into partially and fully occluded annotations. Full occlusions are removed, while data augmentation is applied to the partial occlusions to create more of them. In this way, the robustness of the model is improved. Furthermore, they apply data augmentation using Local Grayscale Transform (LGT) and Random Erasing (Zhong et al., 2020). In particular, LGT ensures that color similarities in the jersey

do not lead to problems. Furthermore, Liu et al. overrepresent IDs with less than twenty images to ensure balance in the training set. They use a model ensemble containing 24 model instances. Finally, common re-identification post-processing methods are applied: augmentation test, re-ranking (Zhong et al., 2017) and query expansion (Chum et al., 2007).

## 6 Results

Statistics on each challenge, as well as the overall winners, are given in Table 1. The segmentation and object detection challenges were the most popular challenges by some margin. We note that, except for in the 2022 edition of the action recognition challenge, the achieved results improve with every new edition, with a significant margin. The fact that state-of-the-art solutions improve so much in a year indicates that progress in computer vision still shows no signs of slowing down.

**Comparison to large-scale settings.** We compare the winning entries in our challenges against the ranking leaders of comparable public large-scale benchmarks. For image classification and action recognition, our data-deficient settings are subsets of the same dataset used for a large-scale benchmark: ImageNet and Kinetics-400, respectively. For the object detection and segmentation challenges we used custom datasets from 2021 onward, while we used subsets of large-scale public datasets in 2020: COCO and Cityscapes, respectively. We therefore compare the winning entry in the 2020 edition of these challenges against the ranking leader in the corresponding challenge in 2020. The only edition of the re-identification challenge used a custom dataset and can therefore not be compared to any public large-scale dataset.

Figure 1 shows the comparisons. We find gaps of around fifteen percentage points in all challenges. There are several possible explanations for these gaps. They may be due to our challenges having less competitors and therefore less competitive solutions. We do however expect that a large part of the gap is due to the difficulty of deep learning in a data-deficient setting without using additional data. We therefore posit that research into data-efficient deep learning is still quite necessary.

## 7 Conclusion

Over the course of four years, we organize the VIPriors challenges, challenging competitors to train computer vision models in a data-deficient setting. We provide more samples than in few-shot learning, but much less than given in large-scale benchmarks. Furthermore, networks had to be trained using only the provided data.

The goal of our challenges is to push progress in data-efficient deep learning, ideally by way of infusing prior knowledge into deep learning models. We can conclude that our challenges show progress in data-deficient deep learning, as winning entries improved in performance over the years. We also show that more research into data-efficient deep learning is necessary, as the winning entries in our data-deficient challenges score around 15 percentage points lower than their counterparts in large-scale benchmarks.

We analyze the methodology of challenge entries to make conclusions about successful methods for data-deficient deep learning. We find that winning entries make heavy use of model ensembling and data augmentation. Furthermore, mixing Transformers and CNNs is a recipe for success in our challenges.

As to our goal of encouraging the use of prior knowledge, we stimulate entries to use prior-based methods by introducing a jury prize for the best prior-based method per challenge. Whether or not the jury prize specifically contributes to competitor's motivation, we can state that our challenges inspired some entries to use (novel) prior-based methods, and that some of these entries performed well in the challenges. Overall, however, methods not specifically designed for data-deficient settings, such as model ensembling and data augmentation, contributed most to success.

**Limitations.** Our challenges are specifically aimed at the setting of supervised training with little data. We rule out using any additional data, and therefore any transfer learning methods. An argument can be made that this is overly restrictive with respect to real-world data-deficient settings, where additional data may be

available. A challenge that does not enforce this rule may be more representative of all possible approaches in a data-deficient setting.

Compared to large-scale benchmarks, the number of competitors in our challenges was relatively small. This may affect the confidence we have in general claims made based on the results of our challenges.

Finally, on the organizational side, there was some evidence that competitors evaded the evaluation limits on CodaLab by registering with multiple accounts. As we could not prove this, we did not address this.

**Future work.** Interestingly, we find that methods based on prior knowledge played only a small role in our challenges. We wonder if this means that the approach of integrating prior knowledge is truly not competitive with cheaply implemented methods such as model ensembling, or if there is another reason why these methods did not shine in our competition. Future work may investigate the efficacy of prior knowledge in deep learning more concretely.

Furthermore, future workshops may organize similar challenges around prior-based methods. We recommend devising an incentive through which prior-based methods are explicitly encouraged. This can be an organizational incentive such as our jury prize or an incentive integrated into the task measure, e.g. by scoring entries by some direct measure of their data efficiency (Hoiem et al., 2021).

With our challenges, we solely addressed limits on data availability. A similar concern is the energy cost of deep learning. Combining both concerns in a single challenge may change what methods are successful. For example, model ensembling is an effective way to improve performance with little effort but places a significant extra cost on the energy used by the model.

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

## A    Appendix

The following pages contain the full rankings for all challenges.

Table 2: Overview of challenge entries for image classification challenge. J indicates jury prize. Bold-faced methods are contributions by the competitors. Due to confusion around the competition deadline in 2021, entries marked with † were awarded special rankings.

| Image classification | | | | |
|---|---|---|---|---|
| Rank | Team | Architectures, backbones | Data augmentation | Methods | Acc. |
| 1 (2022) | **Ma et al.** | SE+PyramidNet, ResNeSt200e, ReXNet,EfficientNet-B8, ConvNeXt-XL* (Liu et al., 2022) | CutMix, AutoAugment (Cubuk et al., 2018), Stubborn Image Augmentation(SIA) | Label smoothing, AdvProp, Random image cropping and patching (RICP), extra training on stubborn images, hard fusion | **78.7** |
| 2 (2022) & J (2022) | Lu et al. | HorNet, ConvNeXt (Liu et al., 2022) | Automix (Liu et al., 2021c) | Cross-decoupled knowledge distillation (Zhao et al., 2022), label smoothing | 77.9 |
| 3 (2022) | Zuo et al. | CoAtNet, TResNet, Resnet50, Resnext50, EdgeNeXt | CutMix, Random erasing, MixUp, AutoAugment | Knowledge distillation between encoders | 77.7 |
| 4 (2022) | Wang et al. | ResNeSt, TResNet, SE-ResNet, ReXNet, ECA-NFNet, ResNet-RS (Bello et al., 2021), Inception-ResNet, RegNet, EfficientNet, MixNet | AutoAugment, MixUp, CutMix, padding | label smoothing, train on larger images, data resampling | 76.8 |
| 5 (2022) | She et al. | ResNeSt, Res2Net, Xception, DPN (Chen et al., 2017), EfficientNet, SENet | AutoAugment, MixUp | label smoothing, train on larger images, hard negative resampling | 75.4 |
| 6 (2022) | Chen et al. | ResNeSt, EfficientNet, ReXNet, RegNetY | AutoAugment, MixUp, CutMix, ColorJitter | label smoothing, train on larger images, Exponential Moving Average on network parameters | 70.8 |
| 1 (2021) | **Sun et al.** | ResNeSt (Zhang et al., 2020a) | AutoAugment (Cubuk et al., 2018), MixUp (Zhang et al., 2018), CutMix (Yun et al., 2019) | BlurPool (Zhang, 2019), stochastic depth (Huang et al., 2016) | **75.5** |
| 2 (2021) | J. Wang et al. | ResNeSt (Zhang et al., 2020a), TResNet (Ridnik et al., 2021), RexNet (Han et al., 2021), RegNet (Radosavovic et al., 2020), Inception-ResNet (Szegedy et al., 2017) | AutoAugment (Cubuk et al., 2018), MixUp (Zhang et al., 2018), CutMix (Yun et al., 2019) | label smoothing (Szegedy et al., 2016), DSB-Focalloss | 75.2 |
| 2 (2021) † | Guo et al. | EfficientNet-b5/b6/b7 (Tan & Le, 2019), DSK-ResNeXt101 (Bruintjes et al., 2021) (Xie et al., 2017), ResNet-152 (He et al., 2015), SEResNet-152 (Xie et al., 2017) | AutoAugment (Cubuk et al., 2018), MixUp (Zhang et al., 2018), CutMix (Yun et al., 2019), random erasing (Zhong et al., 2020) | **Contrastive Regularization**, Mean Teacher (Tarvainen & Valpola, 2017), Symmetric Cross Entropy (Wang et al., 2019c), label smoothing (Szegedy et al., 2016), dropout (Srivastava et al., 2014) | 74.3 |
| 3 (2021) & J (2021) | T. Wang et al. | ResNeSt-101/200 (Zhang et al., 2020a), SEResNeXt-101 (Zhang et al., 2020a) | HorizontalFlip, FiveCrop, TenCrop, label smoothing (Szegedy et al., 2016) | **Iterative Partition-based Invariant Risk Minimization** | 71.6 |
| 1 (2020) | **Sun et al. (2020a)** | EfficientNet-b5, EfficientNet-b6, ResNeSt-101, ResNest-200, DSK-ResNeXt50, DSK-ResNeXt101 | Cutmix | **Dual Selective Kernel** based on SK (Li et al., 2019) with anti-aliasing (Zhang, 2019), center loss (Wen et al., 2016), tree supervision loss inspired by (Wan et al., 2020) | **73.1** |
| 2 (2020) | Luo et al. (2020) | ResNest-101 (Zhang et al., 2020a), TresNet-XL (Ridnik et al., 2021), SEResNeXt-101 (Hu et al., 2018) | AutoAugment (Cubuk et al., 2018), Cutmix (Yun et al., 2019) | combinations of cross-entropy loss and triplet loss (Hermans et al., 2017) as well as ArcFace loss (Deng et al., 2019), label smoothing (Szegedy et al., 2016) | 70.2 |
| 2 (2020) † | Zhao & Wen (2020) | MOCO v2 (He et al., 2020a) (teacher), ResNeXt101 (Xie et al., 2017) (student) | AutoAugment (Cubuk et al., 2018) | label smoothing (Müller et al., 2019), ten crops, model ensembling | 68.8 |
| 3 (2020) | Kim et al. (2020a) | EfficientNet (Tan & Le, 2019) | **Low Significant Bit swapping**, RandAugment (Cubuk et al., 2020a) | **focal cosine loss** inspired by focal (Lin et al., 2017b) and cosine (Barz & Denzler, 2020) losses, Exponential Moving Average, dropout, drop connection, Plurality voting ensemble | 67.3 |
| 4 (2020) | Qingfeng Liu & Lee (2020) | EfficientNet (Tan & Le, 2019) | AutoAugment (Cubuk et al., 2018), MixUp (Zhang et al., 2018) | model ensembling | 66.4 |

Table 3: Overview of challenge entries for object detection challenge. J indicates jury prize. Bold-faced methods are contributions by the competitors. Due to confusion around the competition deadline in 2021, entries marked with † were awarded special rankings.

| | | | **Object detection** | | | |
|---|---|---|---|---|---|---|
| Rank | Team | Architectures, backbones | Data augmentation | Methods | AP (COCO) | AP (DelftBikes) |
| 1 (2023) | **Zhao et al.** | Cascade RCNN (Cai & Vasconcelos, 2018b), Swin T. (Liu et al., 2021a), ConvNeXt (Liu et al., 2022), ConvNeXtV2 (Woo et al., 2023) | Albumentations (Buslaev et al., 2020b), PhotoMetricDistortion, MixUp (Zhang et al., 2018), Auto Augment V2 (Cubuk et al., 2018) | FPN (Lin et al., 2017a), SWA (Izmailov et al., 2018b), **recombined synthetic dataset**, **retraining hard classes** | | **34.5** |
| 2 (2023) & J (2023) | Lu et al. | Scaled-YOLOv4 (Wang et al., 2021a), YOLOv7 (Wang et al., 2023a), YOLOR (Wang et al., 2021b), CBNetv2 (Liang et al., 2021a) | Pre-training: random scaling, random flipping, color jitter; fine-tuning: Mosaic Augmentation (Bochkovskiy et al., 2020), Copy-Paste (Kisantal et al., 2019), Mix-Up (Zhang et al., 2018), Cutout (DeVries & Taylor, 2017) | Model Soups (Wortsman et al., 2022), Weighted Boxes Fusion (Roman Solovyev & Gabruseva, 2021), Test-Time Augmentation (Moshkov et al., 2020), **Image Uncertainty Weighted** | | 33.3 |
| 3 (2023) | Jing et al. | YOLOv7 (Wang et al., 2022), YOLOv8x-p2 (git), YOLOv8x, YOLOv8-p6, Cascade RCNN (Cai & Vasconcelos, 2018b) | Mosaic Augmentation (Bochkovskiy et al., 2020), MixUp (Zhang et al., 2018) | Weighted Boxes Fusion (Roman Solovyev & Gabruseva, 2021), Test-Time Augmentation (Moshkov et al., 2020), horizontal flip testing | | 30.6 |
| 4 (2023) | Wang et al. | YOLOv8 (git) | Mosaic Augmentation (Bochkovskiy et al., 2020), MixUp (Zhang et al., 2018) | SparK (Tian et al., 2023) pre-training, Weighted Boxes Fusion (Roman Solovyev & Gabruseva, 2021), Test-Time Augmentation (Moshkov et al., 2020), horizontal flip testing | | 30.4 |
| 5 (2023) | Sun et al. | YOLOv8 (git) | HSV, rotation, translation, scaling, shearing, flipping, Mosaic Augmentation (Bochkovskiy et al., 2020), MixUp (Zhang et al., 2018), Copy-Paste (Kisantal et al., 2019) | GAM (Zhou et al., 2022), cross-validation | | 29.4 |
| 6 (2023) | *Team fha.ddd* | Faster RCNN (Ren et al., 2016), EfficientNet-V2 (Tan & Le, 2021) | AutoAugment (Cubuk et al., 2018) | | | 26.6 |
| 1 (2022) | **Lu et al.** | YOLOv4(Bochkovskiy et al., 2020), YOLOv7(Wang et al., 2022), YOLOR (Wang et al., 2021c), CBNetv2 (Liang et al., 2021a) | Mosaic (Bochkovskiy et al., 2020), MixUp (Zhang et al., 2018), CopyPaste (Kisantal et al., 2019) | Weighted Boxes Fusion (Roman Solovyev & Gabruseva, 2021), TTA, Model Soups (Wortsman et al., 2022), Image Uncertainty Weighted | | **33.0** |
| 2 (2022) | Xu et al. | Cascade RCNN (Cai & Vasconcelos, 2018b), Swin T. (Liu et al., 2021a), ConvNext (Liu et al., 2022), ResNext (Xie et al., 2017), FPN (Lin et al., 2017a) | AutoAugment (Cubuk et al., 2018), random flip, multi-scale augmentations (Liu et al., 2000) | MoCoV3 (Chen et al., 2021), MoBY (Xie et al., 2021b), Soft-NMS (Bodla et al., 2017), SSFPN(Hong et al., 2021), non-maximum weighted (NMW) (Zhou et al., 2017) | | 32.9 |
| 3 (2022) | J. Zhao et al. | Cascade RCNN (Cai & Vasconcelos, 2018b), Swin T. (Liu et al., 2021a), Convnext (Liu et al., 2022), FPN (Lin et al., 2017a) | Albu, MixUp (Zhang et al., 2018), AutoAugment (Cubuk et al., 2018) | SWA (Izmailov et al., 2018a), Hard classes retraining, Soft-NMS (Bodla et al., 2017), pseudo labeling | | 32.4 |
| 4 (2022) & J (2022) | P. Zhao et al. | Cascade RCNN (Cai & Vasconcelos, 2018b), Swin T. (Liu et al., 2021a), Pyramid ViT (Wang et al., 2021e) | Mosaic (Bochkovskiy et al., 2020), MixUp (Zhang et al., 2018) | SimMIM (Xie et al., 2022), GIOU loss (Union, 2019), Soft-NMS (Bodla et al., 2017) | | 30.9 |
| 1 (2021) | **Lu et al.** | YOLO 4-5 (Bochkovskiy et al., 2020) (Jocher & et. al., 2021) | Mosaic (Bochkovskiy et al., 2020), MixUp (Zhang et al., 2018), random color-jittering | Weighted Boxes Fusion (Roman Solovyev & Gabruseva, 2021) | | **30.5** |
| 1 (2021) † | **Zhang et al.** | Cascade RCNN (Cai & Vasconcelos, 2018b), DCN (Dai et al., 2017) | Multi-scale augmentation | TTA, MoCo v2 (Chen et al., 2020c), Soft-NMS (Bodla et al., 2017), Class-specific IoU thresholds | | 30.4 |
| 2 (2021) | Niu et al. | Swin-T (Liu et al., 2021a) | Hierarchical labeling | FPN (Lin et al., 2017a), Soft-NMS (Bodla et al., 2017), pseudo labeling | | 30.4 |
| 2 (2021) † | Luo et al. | Cascade RCNN (Cai & Vasconcelos, 2018b), DCN (Dai et al., 2017) | Albu, Top-Bottom Cut | GCNet (Cao et al., 2019), SimSiam (Chen & He, 2021), Soft-NMS (Bodla et al., 2017) | | 30.1 |
| 1 (2020) | **Shen et al.** (Shen et al., 2020) | Cascade-RCNN (Cai & Vasconcelos, 2018b), Res2Net-152 (Gao et al., 2019), SeNet154 (Hu et al., 2018), ResNeSt-152 (Zhang et al., 2020a) | **Bbox-jitter**, GridMask (Chen et al., 2020b), MixUp (Zhang et al., 2018) | **infuse global context features**, Weighted Boxes Fusion (Solovyev et al., 2021) | **39.4** | |
| 2 (2020) | Gu et al. (Gu et al., 2020) | Cascade R-CNN (Cai & Vasconcelos, 2018b), ResNet-D (He et al., 2018), FPN (Lin et al., 2017a), DCNv2 (Zhu et al., 2019c) | Albumentations (Buslaev et al., 2020a), AutoAugment (Zoph et al., 2019), Stitchers (Chen et al., 2020d), **Mosaics-SC** based on Mosaic (Bochkovskiy et al., 2020) | Libra R-CNN (Pang et al., 2019), Guided Anchoring (Wang et al., 2019a), Generalized Attention (Zhu et al., 2019b), TSD (Song et al., 2020), model ensembling | 36.6 | |
| 3 (2020) | Luo et al. (Luo & Che, 2020) | Scratch Mask R-CNN (Zhu et al., 2019a), ResNet-101 (He et al., 2015) | Albumentations (Buslaev et al., 2020a) | Soft-NMS (Bodla et al., 2017), extra weight on classification loss | 35.1 | |

Table 4: Overview of challenge entries for segmentation challenge. J indicates jury prize. Bold-faced methods are contributions by the competitors.

| | | | **Segmentation** | | | |
|---|---|---|---|---|---|---|
| Rank | Team | Architectures, backbones | Data augmentation | Methods | mIoU (MiniCity) | mIoU (Basketball) |
| 1 (2023) | **Zhang et al.** | HTC (Chen et al., 2019a), Mask RCNN (He et al., 2017), Swin (Liu et al., 2021a), ResNet (He et al., 2016), FPN (Lin et al., 2017a), CBNet (Liu et al., 2020b) | Geometric (Paschali et al., 2019), color space, sharpness, noise injection, Copy-Paste (Kisantal et al., 2019) | **Orthogonal Uncertainty Representation**, Weighted Boxes Fusion (Roman Solovyev & Gabruseva, 2021), Seesaw loss (Wang et al., 2021d), SWALP (Yang et al., 2019) | | **59.0** |
| 2 (2023) | Lu et al. | HTC (Chen et al., 2019b), BEiTv2-L (Peng et al., 2022), ViT-Adapter (Chen et al., 2022), Internimage (Wang et al., 2023b) | Mosaic Augmentation (Bochkovskiy et al., 2020), Copy-Paste (Kisantal et al., 2019), Mix-Up (Zhang et al., 2018), random brightness, random contrast, random saturation, random scale, random flip, sharpen and overlay, blur, Gaussian noise, grid-mask | GIoU loss (Rezatofighi et al., 2019), Soft NMS (Bodla et al., 2017), **expert network** with SegFormer (Xie et al., 2021a) and SeMask (Jain et al., 2023), Test-Time Augmentation (Moshkov et al., 2020), Model Soups (Wortsman et al., 2022), random scaling by Yunusov et al. (Yunusov et al., 2021) | | 58.2 |
| 3 (2023) | Liang et al. | Mask2Former (Cheng et al., 2022a), YOLOX (Ge et al., 2021), DETR (Carion et al., 2020) | Copy-Paste (Kisantal et al., 2019), rotation, mirroring, cropping, scaling, random brightness, random saturation, random contrast, random color equality, sharpness, random noise, random erasure, local erasure | Weighted Boxes Fusion (Roman Solovyev & Gabruseva, 2021), Seesaw loss (Wang et al., 2021d), SWA (Zhang et al., 2020b), Soft NMS (Bodla et al., 2017) | | 55.2 |
| 4 (2023) & J (2023) | Hsu et al. | HTC (Chen et al., 2019b), Mask Scoring R-CNN (Huang et al., 2019), CB-SwinTransformer-Base (Liu et al., 2020b; 2021a) | Players: RGB curve distortion; other objects: salt-and-pepper noise & brightness variations; GridMask (Chen et al., 2020b) | **Basketball Court Detection**, GroupNorm (Wu & He, 2018), SWA (Zhang et al., 2020b), Model Soups (Wortsman et al., 2022) | | 50.9 |
| 1 (2022) | **Yan et al. (2022)** | HTC (Chen et al., 2019a), CBSwin-T (Liang et al., 2021b) | **TS-DA**, **TS-IP** (Yan et al., 2022) Random scaling, cropping | CBFPN (Liang et al., 2021b) | | **53.1** |
| 2 (2022) | Leng et al. | CBNetV2 (Liang et al., 2021b), Swin Transformer-Large (Liu et al., 2021a) | AutoAugment (Cubuk et al., 2018), ImgAug (Jung, 2018), Copy-Paste (Ghiasi et al., 2021), Horizontal Flip and Multi-scale Training | | | 50.6 |
| 2 (2022) | Lu et al. | HTC (Chen et al., 2019a), CBSwin-T (Liang et al., 2021b) ResNet (He et al., 2015) ConvNeXt (Liu et al., 2022) Swinv2 (Liu et al., 2021a) CBNetv2 (Liang et al., 2021b) | MixUp (Zhang et al., 2018), Mosaic Task-Specific Copy-Paste (Yunusov et al., 2021) Color and geometric transformations | CBFPN (Liang et al., 2021b) Group Normalization (Wu & He, 2018) | | 50.6 |
| 3 (2022) | Zhang et al. | HTC (Chen et al., 2019a), CBSwin-T (Liang et al., 2021b) | Location-aware MixUp, RandAugment (Cubuk et al., 2020a), GridMask (Chen et al., 2020b), Random scaling, CopyPaste (Ghiasi et al., 2021), Multi-scale augmentation | Seesaw Loss (Wang et al., 2021d) SWA (Izmailov et al., 2018b), TTA | | 49.8 |
| 4 (2022) | Cheng et al. | HTC (Chen et al., 2019a), CBSwin-T (Liang et al., 2021b) | RandAugment Copy-Paste (Ghiasi et al., 2021) GridMask | Mask Transfiner (Ke et al., 2022) | | 47.6 |
| 5 (2022) & J (2022) | Cheng et al. (2022b) | ResNet (He et al., 2015) | Random flip and scale jitter | Sparse Instance Activation for Real-Time Instance Segmentation (Cheng et al., 2022b) | | 34.0 |
| 1 (2021) & J (2021) | **Yunusov et al. (2021)** | HTC (Chen et al., 2019a), CBSwin-T (Liang et al., 2021b) | **Location-aware MixUp** (Zhang et al., 2018), RandAugment (Cubuk et al., 2020b), GridMask (Chen et al., 2020b), Random scaling | | | **47.7** |
| 2 (2021) | Yan et al. (2021) | Cascade R-CNN (Cai & Vasconcelos, 2018a), ResNet-101 (He et al., 2015) | Random brightness, color jitter, saturation, sharpening, blurring, noise, pixel shuffle, pixelization, filtering, hue transform | Switchable atrous convs. (Qiao et al., 2020) Group normalization (Wu & He, 2018) | | 40.2 |
| 3 (2021) | Chen et al. | Cascade Mask-RCNN (Cai & Vasconcelos, 2018b), SCNet (Vu et al., 2021), Swin (Liu et al., 2021a) | HorizontalFlip, Random scale and crop | | | 36.6 |
| 4 (2021) | Chen, Zheng | ResNet-50 (He et al., 2015) | Instaboost (Fang et al., 2019) | Seesaw Loss (Wang et al., 2021d), Deformable Convolutions (Dai et al., 2017) | | 18.5 |
| 1 (2020) | **Weitao & Zhibing (2020)** | HRNet (Wang et al., 2019b) | **multi-scale** CutMix (Yun et al., 2019) | | **65.64** | |
| 2 (2020) | Liu et al. (2020a) | HRNetv2 (Wang et al., 2019b) | Augmix (Hendrycks et al., 2020) | Object Contextual Representations (Yuan et al., 2019), Online Hard Example Mining (Shrivastava et al., 2016), Frequency Weighted ensemble | 65.61 | |
| 3 (2020) | Hsu & Ma (2020) | HANet (Choi et al., 2020), ResNeSt (Zhang et al., 2020a) | | **edge-preserving loss**, **pasting augmented crops of rare classes** | 64.4 | |
| 4 (2020) | Yesilkaynak et al. (2020) | **EfficientSeg** using MobileNetV3 blocks (Howard et al., 2019) | random hue, random brightness, non-uniform scaling, random rotation, random flipping | | 58.0 | |
| 5 (2020) | Pytel & Motyka (2020) | UNet (Ronneberger et al., 2015) | **CutMix Sprinkles** based on CutMix (Yun et al., 2019) and Progressive Sprinkles (Pro) | | 43.1 | |

Table 5: Overview of challenge entries for action recognition challenge. J indicates jury prize. Bold-faced methods are contributions by the competitors. Due to confusion around the competition deadline in 2021, entries marked with † were awarded special rankings.

| Rank | Team | Architectures, backbones | Data augmentation | Methods | Acc. (UCF101) | Acc. (KineticsViP) |
|------|------|--------------------------|-------------------|---------|---------------|--------------------|
| | | **Action recognition** | | | | |
| 1 (2022) | **Song et al.** | R(2+1)D (Tran et al., 2018), SlowFast (Feichtenhofer et al., 2019), CSN (Tran et al., 2019), X3D (Feichtenhofer, 2020), TANet (Liu et al., 2020c), Timesformer (Bertasius et al., 2021) | Random flipping, TenCrop | Soft voting | | **71** |
| 2 (2022) | He et al. | SlowFast (Feichtenhofer et al., 2019), Timesformer (Bertasius et al., 2021), TIN (Shao et al., 2020), TPN (Yang et al., 2020), X3D (Feichtenhofer, 2020), Video Swin Transformers (Liu et al., 2021b), R(2+1)D (Tran et al., 2018), DirecFormer (Truong et al., 2022). | AutoAugment (Cubuk et al., 2018), CutMix (Yun et al., 2019) random flip, grayscale, jitter, temporal aug., TenCrop, test-time aug. | Label smoothing | | 69 |
| 3 (2022) & J (2022) | Tan et al. | TSN (Wang et al., 2016), TANet (Liu et al., 2020c), TPN (Yang et al., 2020), SlowFast (Feichtenhofer et al., 2019), CSN (Tran et al., 2019), Video MAE (Tong et al., 2022) | MixUp (Zhang et al., 2018), CutMix (Yun et al., 2019) MoCo (He et al., 2020b), TVL-1 (Zach et al., 2007) | | 59 | |
| 1 (2021) & J (2021) | **Dave et al.** | R3D(Hara et al., 2018), I3D(Carreira & Zisserman, 2017), MViT(Fan et al., 2021) | | TCLR(Dave et al., 2021) | | **74** |
| 1 (2021) † | Wu et al. | TPN(Yang et al., 2020), Slowfast (slow path) (Feichtenhofer et al., 2019) | MixUp (Zhang et al., 2018), CutMix (Yun et al., 2019) | | | 66 |
| 2 (2021) | Gao et al. | Swin (Niu et al., 2021), TPN (Yang et al., 2020), X3D(Feichtenhofer, 2020), R2+1D(Tran et al., 2018), TimesFormer(Bertasius et al., 2021), Slowfast(Feichtenhofer et al., 2019) | | | | 73 |
| 1 (2020) | **Ishan Dave & Shah (2020)** | I3D (Carreira & Zisserman, 2017), C3D (Tran et al., 2015), R3D, R2+1D (Tran et al., 2018) | random crops, horizontal flipping, frame skipping | | **90.8** | |
| 2 (2020) | Chen et al. (2020a) | **modified** C3D (Tran et al., 2015) | corner cropping, horizontal flip | **Temporal Central Difference Convolution**, Rank Pooling (Fernando et al., 2017) | 88.3 | |
| 3 (2020) | Luo & Che (2020) | SlowFast (Feichtenhofer et al., 2019) | center/random crop, horizontal flip, normal/reverse video reproduction | TSM (Lin et al., 2019) | 87.6 | |
| 4 (2020) | Kim et al. (2020b) | SlowFast-50 (Feichtenhofer et al., 2019) | **RandAugment-T** based on RandAugment (Cubuk et al., 2020a), **temporal extensions of** CutOut (DeVries & Taylor, 2017), MixUp (Zhang et al., 2018), CutMix (Yun et al., 2019), random crop, random horizontal flip | model ensembling | 86.0 | |

Table 6: Overview of challenge entries for re-identification challenge. J indicates jury prize. Bold-faced methods are contributions by the competitors.

| | | Re-identification | | | |
|---|---|---|---|---|---|
| Rank | Team | Architectures, backbones | Data augmentation | Methods | Acc. |
| 1 (2021) | **Liu et al.** | ResNet (He et al., 2015), ResNetSt (Zhang et al., 2020a), SE-ResNetXt (Xie et al., 2017) (24 models) | Difficult sample mining (Shrivastava et al., 2016), Random Erasing (Zhong et al., 2020), Local Grayscale Transform, affine transformations, pixel padding, random flip. | Triplet loss (Weinberger & Saul, 2009) and circle loss (Sun et al., 2020b), with augmentation test, re-ranking (Zhong et al., 2017), query expansion (Chum et al., 2007) | **96.5** |
| 2 (2021) & J (2021) | Chen et al. | ResNet-IBN (He et al., 2015), SE ResNet-IBN (Hu et al., 2018) (5 models) | Video temporal mining, Random Erasing (Zhong et al., 2020), pixel padding, random flip | Cross-entropy and triplet loss (Weinberger & Saul, 2009), with augmentation test, re-ranking (Zhong et al., 2017), 6x schedule (He et al., 2019). | 96.4 |
| 3 (2021) | Qi et al. | ResNet-IBN (He et al., 2015), 23OSNet on Stronger Baseline | Random Erasing (Zhong et al., 2020), color jitter, random flip, AutoAugment (Cubuk et al., 2018) | Cross-entropy and triplet loss (Weinberger & Saul, 2009), re-ranking (Zhong et al., 2017), query expansion (Chum et al., 2007) | 94.2 |
| 4 (2021) | Zheng et al. | ResNet-IBN (He et al., 2015) w/ spatial and channel attention | Random Erasing and Patch (Zhong et al., 2020), color jitter, random flip, AutoAugment (Cubuk et al., 2018) | Cross-entropy, triplet loss (Weinberger & Saul, 2009) and circle loss (Sun et al., 2020b) | 84.8 |

