# OpenReview forum: "Data-Efficient Challenges in Visual Inductive Priors: A Retrospective"
_TMLR — Rejected by TMLR_

### Review · Reviewer_WWoK · 2025-06-05

**Summary Of Contributions:**

The manuscript provides a retrospective summary of the VIPriors workshop series during 2020-2023, analyzing challenges in data-deficient deep learning for 4 different computer vision tasks. While the workshops aim to motivate prior-based approaches, it turns out that model ensemble and data augmentation contribute more to the success of data-deficient learning.

**Audience:**

Yes

**Claims And Evidence:**

Yes

**Requested Changes:**

See above.

**Strengths And Weaknesses:**

**Strengths**
- The rule of prohibiting transfer learning makes the challenges a useful testbed for data-deficient learning in resource-limited applications.
- The authors curated datasets that can be useful for future benchmarking.

**Weaknesses**
- The paper merely summarizes existing work and challenge outcomes without proposing new methods or insights. Moreover, the outcomes showcase the effectiveness of model ensemble and data augmentation, which is against the goal of promoting prior knowledge-based methods.
- Although the paper is positioned around "visual inductive priors," it offers little analysis or theoretical grounding for their use or effectiveness.
- There lacks experiments analyzing why prior-based methods underperformed model ensembles and data augmentation.
- Results are reported with little statistical analysis and ablation tests. In addition, it would be beneficial if the comparison of model parameters, and computational overheads (time and space complexities) can be provided.

---

### Review · Reviewer_Cco1 · 2025-06-09

**Summary Of Contributions:**

This paper serves as a retrospective report summarizing the authors' organization of the VIPriors workshop series over four years (2020–2023), which centers on data-deficient challenges for computer vision tasks. The core contribution lies in the systematic documentation of challenge setups, results, and trends across five tasks: image classification, object detection, segmentation, action recognition, and re-identification. The paper highlights two main empirical findings: (1) ensembles of Transformer and CNN architectures yield strong performance under limited data, and (2) heavy data augmentation is commonly adopted and beneficial. The authors argue for the necessity of further research in data-efficient learning and propose ideas for future challenge design.

**Audience:**

No

**Claims And Evidence:**

Yes

**Requested Changes:**

- Clarify the Scope and Nature of the Paper:

    - Explicitly state early on that this is a summary/report paper rather than a novel research contribution. Consider submitting to a workshop or as a community report instead of a regular journal track unless more analytical depth is added.

- Deepen the Analysis:

    - Analyze why Transformer + CNN ensembles perform better. Are there specific properties of these architectures that complement each other? Would an ablation study on ensemble size and diversity offer new insights?

    - Discuss whether different tasks (e.g., classification vs. segmentation) benefit differently from specific architectures or ensemble combinations.

- Propose Future Methodological Directions:

    - Rather than only documenting trends, consider proposing a new method or taxonomy to improve data-efficient learning, such as a hybrid pretraining strategy using synthetic data or task-specific priors.

**Strengths And Weaknesses:**

## Strengths:

- The paper provides a comprehensive summary of multiple years of community-driven benchmarks under a constrained data regime.

- The retrospective may serve as a useful resource for newcomers to data-efficient learning or organizers of similar benchmark challenges.

## Weaknesses:

- Lack of Novelty: The paper mainly serves as a workshop summary and does not constitute an academic research paper with original technical contributions. It lacks new theoretical analysis, methodological innovations, or experimental insights beyond what is already evident from the challenge results.

- Superficial Insights: The main findings — ensembling is useful and data augmentation improves performance — are expected and well-known in low-data settings. There is limited discussion explaining why certain ensemble combinations (e.g., CNN + Transformer) work particularly well or whether different tasks benefit from different ensemble strategies.

- Missed Opportunity for Deeper Analysis: The authors observe performance gaps between models trained from scratch in data-deficient settings and those trained with large-scale data, but do not analyze the root causes of these gaps or propose methods to bridge them.

- Challenge Design Limitation: The decision to disallow transfer learning, while consistent with the workshop’s philosophy, might limit the challenge's relevance to practical scenarios where transfer learning is a standard tool. Moreover, the selected tasks do not reflect common real-world data-deficient applications such as medical imaging or remote sensing.

---

### Review · Reviewer_cf2J · 2025-06-20

**Summary Of Contributions:**

This paper details the Visual Inductive Priors (VIPriors) workshop series, which ran from 2020 to 2023, focusing on the challenging problem of training deep learning models with limited data in computer vision. A key contribution is the establishment of challenges that uniquely forbid transfer learning, forcing participants to train models from scratch on small datasets. The new knowledge presented includes insights into successful strategies in data-deficient settings.

**Audience:**

Yes

**Broader Impact Concerns:**

The paper notes that deep learning depends on large-scale datasets and training, contributing to carbon emissions. While the VIPriors challenges aim for data efficiency, the successful entries often rely on "large model ensembles" and "heavy use of data augmentation". Ensembling, in particular, can significantly increase the computational resources and energy required for deployment and inference, even if individual models are data-efficient. A broader impact statement could discuss this trade-off: while data efficiency is good, the reliance on large ensembles might negate some environmental benefits, contributing to increased energy consumption and carbon footprint in practice.

**Claims And Evidence:**

Yes

**Requested Changes:**

Critical to address because it directly impacts the reliability and validity of the observed challenge - The authors acknowledge evidence of competitors evading evaluation limits by registering with multiple accounts. While they state they "cannot prove this" and "do not address this" , this compromises the fairness and integrity of the challenge results, especially concerning overfitting to the test set. For acceptance, the authors should propose a concrete plan or discuss potential strategies implemented in future iterations to detect and deter such behavior. This might involve more robust registration verification, IP address tracking, or a stricter penalty system.

**Strengths And Weaknesses:**

Strengths:

(1)  The VIPriors challenges are unique in explicitly prohibiting transfer learning and requiring training from scratch in data-deficient settings. This design effectively pushes the boundaries of research in data-efficient deep learning.(2) The paper provides a valuable retrospective of four years of challenges across five distinct computer vision tasks (image classification, object detection, segmentation, action recognition, and re-identification). This offers a broad overview of trends and successful methods. (3) The analysis clearly identifies that large model ensembles and heavy data augmentation are key factors in successful entries in data-deficient settings. The observation that hybrid CNN-Transformer architectures are prevalent is also insightful. (4) Finally, the paper effectively demonstrates the significant performance gap (around 15 percentage points) between data-deficient and large-scale benchmarks. This gap strongly justifies the continued need for research in data-efficient deep learning.

Weakness:  (1) While a core aim was to stimulate novel prior-based methods, the paper notes that these were not widely adopted and often succeeded only in conjunction with more established techniques like ensembling and data augmentation. The authors speculate this might be due to the effort involved. This aspect could be further explored or re-emphasized in the discussion regarding the primary objective. (2) For challenges with fewer entries, like action recognition and re-identification, the authors admit it's hard to draw strong conclusions. While understandable, perhaps a brief discussion on potential reasons for lower interest in these specific tasks (e.g., complexity of datasets, niche application) could add more value.

---

### Review · Reviewer_Kgeh · 2025-06-21

**Summary Of Contributions:**

The contribution of this paper is a detailed analysis of the performance and methodologies employed by competitors in these challenges, all of which focused on training deep learning models for computer vision tasks with limited data and without allowing transfer learning from pre-trained models.

**Audience:**

Yes

**Claims And Evidence:**

Yes

**Requested Changes:**

For action recognition and re-identification, where general trends are hard to confirm due to fewer entries, a more nuanced qualitative discussion could be provided if quantitative analysis is not feasible. Perhaps discussing unique aspects of the few successful entries in these categories, or what made them distinct from other, less successful approaches.

**Strengths And Weaknesses:**

strength:
1. The paper offers a thorough and structured overview of four years of VIPriors challenges, covering various computer vision tasks and their evolution over time.
2. It directly addresses a critical and relevant problem in deep learning: training models with limited data. The strict rule against transfer learning or pre-training from other datasets makes the findings particularly relevant for truly data-deficient scenarios.

Weakness:
1.  For action recognition, it's hard to confirm general trends due to fewer entries, which somewhat limits the generalizability of some findings.
2. For Re-ID task, the methods this paper used are from the same year 2021, which can indicate a bias.

---

### Author Response · Authors · 2025-06-26
**Rebuttal**

We thank the reviewers [Kgeh, cf2J, Cco1W, WWoK] for their extensive effort in reviewing our work.

First, we appreciate the **praise** of our work: reviewers judge our work as “thorough” [Kgeh] and “comprehensive” [Cco1], a “structured” [Kgeh] and “broad” [cf2J] overview, and as a “useful resource” [Cco1]. The choice to prohibit transfer learning in data-deficient settings makes it “a useful testbed” [WWoK], “particularly relevant for truly data-deficient scenarios” [Kgeh] and “pushes the boundaries of research in data-efficient deep learning” [cf2J]. Our work “clearly identifies” [cf2J] key factors in successful data-deficient methods. It also “effectively demonstrates the significant performance gap [...] between data-deficient and large-scale benchmarks” [cf2J], which “justifies the continued need for research in data-efficient deep learning” [cf2J].

While our submission does not contribute novel methodology, it falls within TMLR’s scope of survey and community papers. Our key contributions are the systematic reporting of results, evidence of practical efficacy of known methods, and insights into the behavior of methods in data-deficient settings. We will discuss the comments of reviewers on these topics below.



**Scope**

> The paper mainly serves as a workshop summary and does not constitute an academic research paper with original technical contributions. It lacks new theoretical analysis, methodological innovations, or experimental insights beyond what is already evident from the challenge results. [...] Rather than only documenting trends, consider proposing a new method or taxonomy to improve data-efficient learning, such as a hybrid pretraining strategy using synthetic data or task-specific priors. [...] Consider submitting to a workshop or as a community report instead of a regular journal track unless more analytical depth is added. – [Cco1]

> The paper merely summarizes existing work and challenge outcomes without proposing new methods or insights. – [WWoK]

We understand that reviewers Cco1 and WWoK expect a TMLR paper to have a methodological contribution. However, we note that TMLR explicitly lists “accounts of applications of existing techniques that shed light on the strengths and weaknesses of the methods” and “surveys that draw new connections, highlight trends, and suggest new problems in an area” as in-scope for TMLR [1]. We believe our work is a valuable contribution to the field, even if it does not submit any novel methodology.

> Explicitly state early on that this is a summary/report paper rather than a novel research contribution. – [Cco1]

Agreed. We will more clearly mention this in the introduction.

**Novelty of findings**

> The main findings — ensembling is useful and data augmentation improves performance — are expected and well-known in low-data settings. – [Cco1]

We agree that the effectiveness of augmentation and ensembling is known in principle. It was, however, not known beforehand that such techniques consistently dominate in a competitive data-deficient benchmark when compared to other methods. The fact that they win challenges confirms, for the first time, their real-world importance and reproducibility.

_Continued in next post..._

---

> ### Author Response · Authors · 2025-06-26
> **Rebuttal (cont'd)**
>
> _Continued from previous post._
>
> **Missing theoretical grounding**
>
> > Although the paper is positioned around "visual inductive priors," it offers little analysis or theoretical grounding for their use or effectiveness. – [WWoK]
>
> Thank you for the comment. We believe several empirical works [2-4] as well as theoretical works [5-6] support the argument of the effectiveness of visual inductive priors. We will add these references to the paper.
>
> **Fighting cheating**
>
> > For acceptance, the authors should propose a concrete plan or discuss potential strategies implemented in future iterations to detect and deter such behavior. This might involve more robust registration verification, IP address tracking, or a stricter penalty system. – [cf2J]
>
> We relied on CodaLab to host the challenges, and therefore on their security system. Unfortunately, CodaLab does not seem to be actively developed to improve these issues. We would recommend future competition organizers to consider alternative systems with better security. We will include this in the paper.
>
> **Impact on compute**
>
> > A broader impact statement could discuss this trade-off: while data efficiency is good, the reliance on large ensembles might negate some environmental benefits, contributing to increased energy consumption and carbon footprint in practice. – [cf2J]
>
> Thank you for raising this issue. We agree wholeheartedly, and will include a note in the discussion of the paper.
>
>
> Finally, we would like to thank the reviewers again for raising important issues and making valuable suggestions to improve this work. We will revise the paper using these suggestions, and hope that this discussion improves your confidence in this work.
>
>
>
> **References**
>
> [1] https://jmlr.org/tmlr/editorial-policies.html
> [2] R. Zhang. “Making Convolutional Networks Shift-Invariant Again”. In: CoRR abs/1904.11486 (2019). arXiv: 1904.11486. URL: http://arxiv.org/abs/1904.11486.
> [3] O. S. Kayhan and J. C. v. Gemert. “On translation invariance in cnns: Convolutional layers can exploit absolute spatial location”. In: Proceedings of the IEEE/CVF Conference on Computer Vision and Pattern Recognition. 2020, pp. 14274–14285.
> [4] Brigato, L., Barz, B., Iocchi, L., & Denzler, J. (2021). Tune it or don't use it: Benchmarking data-efficient image classification. In Proceedings of the IEEE/CVF international conference on computer vision (pp. 1071-1080).
> [5] Li, Z., Zhang, Y., & Arora, S. (2020). Why are convolutional nets more sample-efficient than fully-connected nets?. arXiv preprint arXiv:2010.08515.
> [6] Wang, Z., & Wu, L. (2023). Theoretical analysis of the inductive biases in deep convolutional networks. Advances in Neural Information Processing Systems, 36, 74289-74338.

---

> ### Author Response · Authors · 2025-06-26
> **Rebuttal (cont'd)**
>
> _This is the second part. Conclusion is in the previous (meant to be third) part. Sorry for the confusion._
>
> **Lack of experiments/analysis**
>
> > The authors observe performance gaps between models trained from scratch in data-deficient settings and those trained with large-scale data, but do not analyze the root causes of these gaps or propose methods to bridge them. – [Cco1]
>
> The root cause of the lower performance in data-deficient settings is known, and something we introduce deliberately: the lack of data.
> In our paper we survey data-efficient methods, and thus we contribute to understanding of data-efficient methods, so that others can propose new methods that bridge this gap.
>
> > There lacks experiments analyzing why prior-based methods underperformed model ensembles and data augmentation. – [WWoK]
>
> We believe it is an oversimplification to state that prior-based methods were outperformed by ensembles and augmentations. Rather, prior-based methods had success in our competition, but ensembles and augmentations did as well and were used more often. Prior-based methods can be effective, but their efficacy depends on their implementation, which makes general claims about them difficult. We will update the paper to reflect this position.
>
> We think that prior-based methods are used less because they are generally more difficult and time-consuming to implement. We surveyed the participants on why they chose their particular methods. Unfortunately, we did not get enough responses to do any meaningful analysis, so this was not included in the paper. We will include a note about this survey in the paper.
>
> > Results are reported with little statistical analysis and ablation tests. In addition, it would be beneficial if the comparison of model parameters, and computational overheads (time and space complexities) can be provided. – [WWoK]
>
> We completely agree. Yet, ss we do not have access to the models used in the competition submission, we cannot, unfortunately, independently test the solutions. We can only report the results that the competitors submitted in their technical reports. For future challenges, we recommend setting standards for the reporting of such details. We will include this recommendation in the paper.
>
> > For challenges with fewer entries, like action recognition and re-identification, the authors admit it's hard to draw strong conclusions. While understandable, perhaps a brief discussion on potential reasons for lower interest in these specific tasks (e.g., complexity of datasets, niche application) could add more value. – [cf2J]
>
> Yes! For both tasks, the limited number of submissions was partly caused by our own decision not to host the challenges at all editions. The complexity of each task’s setup and the higher compute requirements for these tasks compared to the other tasks likely also played a role. We will include this brief discussion in the work.

---

### Decision · Action_Editor_NaVo · 2025-08-04

**Recommendation:** Reject

**Audience:**

Yes

**Audience Explanation:**

This paper mostly summarise statistics over methods use to solve some deep learning challenge but don't go very deep in the details and insight to understand what are the keys factor.

**Claims And Evidence:**

No

**Claims Explanation:**

While survey papers can be valuable for the community, especially when they attempt to consolidate and organize insights across past Data-Efficient challenges, I find that this paper falls short of the expectations for a TMLR survey

In particular, although the paper summarizes which methods led to strong results in various benchmark challenges, it lacks the analytical depth needed to extract meaningful or novel insights. For example, when discussing hybrid architectures or model ensembling, the paper tends to group techniques under broad categories without critically examining how or why certain architectural choices contributed to better performance, or what trade-offs they introduced.
As Reviewer WWok and Reviewer Cco1  mention the paper "merely summarizes past challenges with no significant insights", I ten dto align with reviewer observation.

The paper does not seem to identify new patterns or trends that go beyond what is already known in the literature or among practitioners. It remains largely descriptive, reiterating known approach without offering a synthesis that could help the community understand underlying principles or emerging directions.

The paper could be improved with deeper analysis that will bring new insight on the underlying success of the best performing method.
I encourage the author to incorporate the reviewer feedback in their submission in order to make the paper stronger

**Resubmission Of Major Revision:**

The authors may consider submitting a major revision at a later time.